# Using an Antibiogram Profile to Improve Infection Control and Rational Antimicrobial Therapy in an Urban Hospital in The Gambia, Strategies and Lessons for Low- and Middle-Income Countries

**DOI:** 10.3390/antibiotics12040790

**Published:** 2023-04-21

**Authors:** Saffiatou Darboe, Ruel Mirasol, Babapelumi Adejuyigbe, Abdul Khalie Muhammad, Behzad Nadjm, Annabelle De St. Maurice, Tiffany L. Dogan, Buntung Ceesay, Solomon Umukoro, Uduak Okomo, Davis Nwakanma, Anna Roca, Ousman Secka, Karen Forrest, Omai B. Garner

**Affiliations:** 1Medical Research Council Unit, The Gambia at the London School of Hygiene and Tropical Medicine, Banjul P.O. Box 273, The Gambia; 2Department of Pathology and Laboratory Medicine, University of California Los Angeles, Los Angeles, CA 90095, USA; 3David Geffen School of Medicine, University of California, UCLA, Los Angeles, CA 90095, USA; 4University College London Hospital NHS Foundation Trust, London NW1 2BU, UK; 5Department of Pediatrics, Division of Infectious Diseases, David Geffen School of Medicine, UCLA, Los Angeles, CA 90095, USA; 6Department of Clinical Epidemiology and Infection Prevention, University of California, UCLA Health, Los Angeles, CA 90095, USA

**Keywords:** cumulative antibiogram, antimicrobial resistance (AMR), infection prevention and control (IPC), low- and middle-income countries (LMICs), *Escherichia coli* (*E. coli*), *Staphylococcus aureus* (*S. aureus*), *Klebsiella pnuemonaie* (*K. pneumoniae*), bacteraemia, urinary tract infection (UTI)

## Abstract

Antimicrobial resistance is a global health threat and efforts to mitigate it is warranted, thus the need for local antibiograms to improve stewardship. This study highlights the process that was used to develop an antibiogram to monitor resistance at a secondary-level health facility to aid empirical clinical decision making in a sub-Saharan African county. This retrospective cross-sectional descriptive study used 3 years of cumulative data from January 2016 to December 2018. Phenotypic data was manually imputed into WHONET and the cumulative antibiogram constructed using standardized methodologies according to CLSI M39-A4 guidelines. Pathogens were identified by standard manual microbiological methods and antimicrobial susceptibility testing was performed using Kirby-Bauer disc diffusion method according to CLSI M100 guidelines. A total of 14,776 non-duplicate samples were processed of which 1163 (7.9%) were positive for clinically significant pathogens. Among the 1163 pathogens, *E*. *coli* (*n* = 315) *S. aureus* (*n* = 232), and *K. pneumoniae* (*n* = 96) were the leading cause of disease. Overall, the susceptibility for *E. coli* and *K. pneumoniae* from all samples were: trimethoprim-sulfamethoxazole (17% and 28%), tetracycline (26% and 33%), gentamicin (72% and 46%), chloramphenicol (76 and 60%), and ciprofloxacin (69% and 59%), and amoxicillin/clavulanic (77% and 54%) respectively. Extended spectrum beta-lactamase (ESBL) resistance was present in 23% (71/315) vs. 35% (34/96) respectively. *S. aureus* susceptibility for methicillin was 99%. This antibiogram has shown that improvement in combination therapy is warranted in The Gambia.

## 1. Introduction

Antimicrobial resistance (AMR) is recognized as a global health threat requiring appropriate containment strategies particularly with sub-Saharan Africa (sSA) disproportionately affected by this phenomenon compared to other regions of the globe [1,2]. Multiple factors transcending disciplines contribute to the development of AMR, with inappropriate use of antibiotics regarded as a major contributing factor according to the report by the WHO Global Action Plan (GAP) on antimicrobial resistance [3]. Surveillance, one of the pillars of the WHO GAP, is key in understanding the epidemiology of AMR to inform appropriate public health intervention and control is suboptimal in LMICs [4]. Data from the Global North has shown the important role surveillance plays in highlighting resistance trends with the recent GLASS report including consumption data [5]. In contrast, data from sSA is scarce partly due to underutilization of available data and other challenges [6]. Efforts to optimize use of data and improve antimicrobial stewardship from sSA is warranted to combat antimicrobial resistance, thus, improving antimicrobial use as prescribed in the WHO essential Medicines List using the Access, Aware, and Reserve (AWaRe) classification [7]. Surveillance is paramount in monitoring resistance trends and can be an effective, data driven strategy for evidence-based decision making on antimicrobial stewardship intervention [5]. 

The cumulative antibiogram periodically summarizes a healthcare facility’s antimicrobial susceptibility profiles for public health surveillance to identify changes in trends and aid clinicians in making informed empiric therapeutic choices relative to local context [8,9]. Furthermore, antibiograms are useful in detecting potential infectious disease outbreaks [10]. The use of such aggregate data on local or regional resistance trends is fundamental to discern differences and changes in patterns for appropriate selection of antimicrobials for rational use and epidemiological surveillance [1,10,11]. Thus, diagnostics remain key in combating AMR and is important that laboratory staff are aware of the public health and clinical significance of their findings for appropriate action [10,12]. Essential containment strategies including monitoring antibiotic usage, and resistance patterns, using appropriate guidelines for treatment, and establishing effective infection prevention and control (IPC) to reduce transmission of multi-drug resistant organisms and blood culture contamination are essential. Implementing an antibiogram to improve antimicrobial policy will require a substantial commitment on the part of the clinical, educational, and administrative staff; thus, requiring institutional commitment. The inclusion and involvement of physicians in the process is useful in understanding of the interpretation of the antibiogram to facilitate implementation in routine prescribing as recommended [13]. 

In low- and middle-income countries (LMIC) including The Gambia, major gaps exist ranging from limited expertise, infrastructure, investment, and lack of institutional microbiologic diagnostic capabilities [14,15]. In addition, many countries in sSA have not prioritized AMR preparedness and implementation of the global action plan to tackle AMR [6,16]. Furthermore, IPC measures to reduce hospital associated infections and antimicrobial resistance surveillance are fundamental for an integrated health system strengthening approach. The understanding of the clinical implications of resistance patterns and interpretation of an antibiogram is paramount for clinical decision making. Thus, the diagnostic microbiology lab must ensure that the antibiogram is generated in collaboration with physicians, pharmacists, and other infectious diseases professionals for improved antimicrobial stewardship [17]. Additionally, antibiogram data must be generated from quality-assured clinical diagnostic microbiology laboratories using standardized methodologies such as the Clinical Laboratory Standards Institute (CLSI) M39-A4 consensus document [18] to ensure accuracy and reliability for better patient outcomes. With the global spread of antimicrobial resistance [2,19], it is a useful tool to monitor local changes to provide evidence-base data for local prescribing guidelines. However, it is important that ways of disseminating the information to all prescribers are facilitated for easy access in prescribing areas. It is also important that users are educated in its effective use and effect on patient outcome [20] and thus this study facilitated that in a local hospital. Considering this, the Medical Research Council Unit The Gambia at LSTHM (MRCGatLSHTM) and University of California, Los Angeles (UCLA) formed a working collaborative group to support the microbiological diagnostic capabilities and implementation of an antibiogram to aid empiric therapy and IPC to improve antimicrobial stewardship in The Gambia. This project has provided evidence for the feasibility of harnessing collaborative technical support for implementing AMR stewardship in a sub-Saharan African country. 

## 2. Results

### 2.1. Samples and Pathogen Distribution

Over the 3-year period, a total of 14,776 different specimens were received and processed (Figure 1, flowchart), of which 1163 clinically significant organisms were isolated. Urine and blood cultures accounted for one-third (*n* = 4914) and (*n* = 4382) of the samples respectively.

Results of 182 (4.2%) clinically significant blood cultures (Figure 2a) were further analyzed and the leading cause of bacteremia were *E. coli* (*n* = 40) and *S. aureus* (*n* = 39) responsible for 22.0% and 21.4% respectively (Figure 2b). This was followed by non-typhoidal *Salmonella* (NTS) (*n* = 16), *S. pneumoniae* (*n* = 15), *Enterococcus* species (*n* = 15), and *Pseudomonas* species other than *P. aeruginosa* (*n* = 10) (Table 1). From urine samples, 9.0% were confirmed clinically relevant (*n* = 442) (Figure 2a). Among the causes of UTI, 55.4% were *E. coli* (*n* = 245) and 12.7% were *K. pneumoniae* (*n* = 56) (Figure 2b) followed by other coliform bacteria (*n* = 31), *Candida* species (*n* = 26), *Enterococcus faecalis n* = 16 and *Streptococcus agalactiae n* = 14 (Table 2).

Out of 2821 swabs, a total of 381 (13.5%) were considered clinically significant. The swabs were further stratified into ear, eye, ear, nose & throat (EENT) *n* = 1628, skin/wound *n* = 1131 and urogenital *n* = 62 (Figure 1). When stratified according to swab type, 1.2% pathogens were recovered from EENT (*n* = 19), 3.2% of pathogens were recovered from skin/wound (*n* = 355) and 11.3% from urogenital swabs (*n* = 7). The leading pathogens from EENT were *H. influenzae* non-type b (*n* = 5), *K. pneumoniae* (*n* = 4) and *S. pneumoniae* (*n* = 4) (Appendix A). For skin/wound swab, the leading pathogens were *S. aureus* (*n* = 148), *Pseudomonas aeruginosa* (*n* = 39) and *S. pyogenes* (*n* = 38). From urogenital swabs, *S. agalactiae* (4/7) and *N. gonorrhoea* (3/7) were recovered. Among the 390 aspirates, 12.6% (*n* = 49) were considered clinically significant. They were stratified into abscess (*n* = 25) and aspirates from invasive sterile sites (*n* = 365). Pathogenic organisms were recovered from 96% abscesses (24/25) and 6.8% of invasive aspirates (25/365). The organisms from abscesses were 83.3% *S. aureus* (*n* = 20), one each of *Morganella morganii*, NTS, *Streptococcus* group F and *K. pneumoniae.* Among 25 pathogens from invasive sterile aspirates, *Streptococci* species (*n* = 7) and *S. aureus* (*n* = 7) were most prevalent, followed by other coliform bacteria (*n* = 4), two *S. pneumoniae* and two *Pseudomonas* species (Appendix A). Among 334 sputa samples, 26.0% were positive for a pathogen (*n* = 87). The leading pathogens were *H. influenzae* non-type b *n* = 20 (23%) and *K. pneumoniae n* = 17 (19.5%) (Figure 2b). Over the time period studied, only (1.7%) of 353 CSF grew pathogens (*n* = 6) of which four were *S. pneumoniae*, one *E. coli* and one *Pseudomonas* species. Among the stool samples, 13 pathogens were isolated of which (*n* = 10) were NTS.

Overall, *E. coli* (27.1% of the pathogens)and *S. aureus* (19.9%) and were found to be the leading cause of disease in our setting (Figure 2b) followed by *Klebsiella pneumoniae* 8.3% (*n* = 96), *Pseudomonas aeruginosa* (*n* = 57), other coliform bacteria (*n* = 52), *Streptococcus pyogenes* (*n* = 45), *Enterococcus* species (*n* = 43), *Pseudomonas* species other than *aeruginosa* (*n* = 43), *Proteus* species (*n* = 41), *Streptococcus pneumoniae* (*n* = 31) and *Klebsiella* species other than *pneumoniae* (*n* = 30), non-typhoidal *Salmonella* (*n* = 28) and *Streptococcus agalactiae* (*n* = 27) (Figure 2b). *E coli* was prevalent in urine, blood and swabs whilst *S. aureus* was prevalent in swabs, blood and aspirates. In addition, *K. pneumoniae* was prevalent in urine, sputum and swabs.

### 2.2. Susceptibility Profiles of Pathogens

Susceptibility patterns for the common pathogens found in blood and/or urinary pathogens for various antimicrobials was varied (Table 1 for blood culture and Table 2 for urinary pathogens).

The major causative agents for UTI were highly susceptible to nitrofurantoin (94%). Overall, the *Enterobacterales* (except NTS) susceptibility to ampicillin was 11% for *E. coli* and 44% for *Proteus* species. Susceptibilities to 3rd generation cephalosporins for *E. coli* and *K. pneumoniae* were 86% and 54%, for cephamycin 97.8% and 97.9% and for the beta-lactamase inhibitor clavulanic acid 77.2% and 65% respectively. ESBL resistance was present 23% (71/315) vs. 35% (34/96) for *E. coli* and *K. pneumoniae* respectively. Susceptibilities to chloramphenicol, ciprofloxacin and gentamicin for *E. coli* were 75.7%, 72.7% and 75.9% respectively. For *K. pneumoniae* 60%, 63.5%, 53.2% and for *S. aureus* 97.3%, 98.9%, 95.2% respectively (Figure 3). Remarkably, susceptibility for methicillin for *S. aureus* was very high 99.4% but low for penicillin 20.2%. *S. pneumoniae* and *H. influenzae* were susceptible to penicillin 77.4% and ampicillin 100% respectively but low for trimethoprim-sulfamethoxazole (0%). Penicillin susceptibility for *Enterococcus* was 81.4%. *S. agalactiae* and *S. pyogenes* susceptibilities were 100% susceptible for penicillin but for tetracycline 33.3% and 29.6% respectively. These susceptibility profiles are shown in the heatmap depicting increasing resistance with from purple to red (Figure 3).

## 3. Discussion

This study provides an antibiogram profile from the diagnostic microbiology laboratory for clinical samples over a three-year period in The Gambia. For the first time we have analyzed our antibiogram using the appropriate CLSI guideline to incorporate into our local antimicrobial prescription to aid in empirical clinical decision making and have shown that WHO access antimicrobials remain appropriate in our setting albeit with some noted increasing resistance patterns. Key in our findings is the appropriateness of methicillin for *S. aureus* infections and nitrofurantoin for our UTIs. The study has confirmed *E. coli* and *S. aureus* as the most prevalent cause of all disease and in particular bacteremia. It is important to note that *S. aureus* remains a prevalent cause of bacteremia [21,22] as seen in other parts of globe [23,24]. However, *E. coli* is for the first time being reported in the top two bacteremia pathogens, taking over from *S. pneumoniae* as reported previously [21,22]. In addition, it remains the prevalent cause of UTIs. The reduced importance of *S. pneumoniae* as a cause of bacteremia in The Gambia is attributed to an increased proportion of children receiving pneumococcal conjugate vaccine the increasing impact of pneumococcal conjugate vaccines over the years after a higher prevalence of children have been vaccinated [25,26]. The vaccine effectiveness studies done in The Gambia have shown both substantial reduction in incidence of pneumococcal bacteremia among young children [25] and in pneumonia hospitalization [27]. 

Invasive NTS remains a cause of bacteremia at par with *S. pneumoniae* and *Enterococcus* species, albeit with low resistance found in this setting. Notwithstanding, the geographical differences noted for NTS serovars and resistance patterns in previous studies [28,29,30] warrant advances in surveillance and monitoring to provide information on prescribing. The coagulase negative *Staphylococci* and viridan group *Streptococci* were mainly considered contaminant unless in patients with underlying conditions such as infective endocarditis, malignancy and neutropenia after meeting criteria for clinical relevance [31]. It is particularly important to state that quality improvement to reduce contaminant rates and improved collection techniques to increase yield in blood cultures is also ongoing in our facility. IPC interventions targeted to reduce blood culture contamination include standardizing policies, procedures, and training, performing competency checks, and monitoring trends in contamination rates. Initiating a hand hygiene observation process and understanding the cultural implications of blood collection reluctance in patients were additional approaches taken to establish an effective IPC program. With the global spread of antimicrobial resistance [2,19], implementing this antibiogram is a useful tool to monitor local changes to provide evidence-base data for our local prescribing guidelines. The significance of this exercise was the inclusion of relevant stakeholders that supported the disseminating of the information to all prescribers and facilitated easy access in prescribing areas. Users were educated in its effective use and effect on patient outcome [20] and thus this approach could be replicated in similar settings with diagnostics facilities in local hospitals.

The urine samples harbored the most common bacterial isolates with *E. coli*, *K. pneumoniae* and other coliforms as the leading cause of community acquired UTI. Patients were not stratified by age, sex, symptoms or with co-morbidities as this was beyond the scope of this study. A more robust study considering these factors is warranted considering it’s a significant cause of morbidity and infection burden globally [32,33]. Another prospective study just concluded in a different facility investigating risk factors for UTIs did confirm *E. coli* as the most prevalent and over the counter medications as important risk factor (Kebbeh et al. unpublished data). It is important to note that the use of indwelling catheters is uncommon in this setting. However, the overall causes of UTIs in The Gambia are of Gram-negative origin as in other studies [32,34,35]. *Enterococcus faecalis* and *Streptococcus agalactiae* was also found to cause disease. 

The susceptibilities for *E. coli* were similar for both disease syndromes except for ciprofloxacin which was comparatively lower for urinary tract isolates (69% vs. 85%) than for bacteremia isolates. This warrants further investigation into the genomic epidemiology of this important invasive pathogen. Although susceptibility profiles for *E. coli* and other *Enterobacterales* remain high except for *K. pneumoniae*, lower susceptibility is reported in this study than previously for ciprofloxacin, gentamicin, chloramphenicol, and 3rd generation cephalosporins [22]. It is important to highlight that this is first time resistance reports of more than 20% is being reported for chloramphenicol, ciprofloxacin and gentamicin for *E. coli* in this setting. The prevalence of extended spectrum beta-lactamase in 23% and 35% for *E. coli* and *K. pneumoniae* suggests emerging resistance, thus warrants surveillance for appropriate intervention. In addition, the lower ciprofloxacin susceptibility profile seen for *E. coli* in UTI compared to bacteremia deserves further investigation and characterization. Multidrug resistance was evident especially for ampicillin, tetracycline and trimethoprim-sulfamethoxazole as previously seen. Both Gram-negative and Gram-positive pathogens had low susceptibilities profiles for trimethoprim-sulfamethoxazole and tetracycline making these ineffective in our setting. It is worth highlighting that carbapenem resistance is yet to be established routinely. Importantly, in the era of the decline of pneumococcal meningitis and bacteremia [22,25,27], we continue to find increasing resistance to penicillin in this study highlighting the need for surveillance. The WHO recommended empiric drug of choice for sepsis remain for ampicillin and gentamicin. This data therefore provides evidence for this combination’s modification to include cloxacillin and amikacin. *S. aureus* remains susceptible to cefoxitin, the proxy for methicillin and show that cloxacillin remains effective in our setting as previously described [22,33,36]. Penicillin, tetracycline, and trimethoprim-sulfamethoxazole susceptibility however was low for most pathogens, thus highlighting the need for their exclusion in empirical treatment. In addition, despite low HIV prevalence in The Gambia, trimethoprim-sulfamethoxazole has been used as pneumocystis prophylaxis and penicillin in patients with sickle cell syndrome. Moreover, these drugs are widely available to the public and misused due to lack of regulation. 

In the advent of AMR, data on local context should be used to inform therapeutic guidelines and improve prescribing and impact on stewardship. To overcome the challenges of AMR, this study employed a combination of strategies including improving sanitation and IPC to reduce hospital acquired infections and the spread of resistant organisms, episodes, and the use of guidelines to standardized diagnostics and therapeutics. The emerging penicillin resistance warrants further surveillance and highlights the need for improved microbiologic diagnostic capabilities and local antibiogram for appropriate antimicrobials. In resource-limited settings where there has been an increase in empiric treatment with WHO ‘reserve’ antibiotics [7], the knowledge and use of a local antibiogram enables the use of alternative ‘watch’ and ‘access’ class antibiotics, sometimes in combination. At this facility, for example, guidelines suggest nitrofurantoin for uncomplicated UTI and chloramphenicol with gentamicin for intra-abdominal infection and with penicillin for respiratory infection. Thus, use of ceftriaxone, a watch class antimicrobial is limited, whilst empirical use of carbapenems and other reserve antibiotics can be avoided with some confidence as aggregate data from the lab has shown. This is made possible by the facility’s structured approach to clinical governance and quality improvement, which includes the provision of a set of locally written clinical guidelines, which are regularly reviewed and easily accessible to prescribers, as well as audits to assess implementation of the guidance. This enables rapid dissemination of the recommended changes to those who need to change their prescribing patterns. This is further reinforced by regular team meetings and handovers which allow senior staff to identify and correct failures to prescriptions in accordance with the guidelines. Hence policy change and implementation were achieved. 

There are several limitations in this study that may hinder its general applicability. First, although the first isolates for patient was included in the analysis, it did not reveal if sample was collected at point of admission or during admission. Hence, we could not reliably identify a potential hospital associated infection. Second, the data was not stratified by age and risk factors. Thirdly, time of sample collection was not documented. Fourth, the data precedes the COVID-19 pandemic and antimicrobial resistance may have increased due to high use. Although no changes have been ascertained, post-pandemic data is being analyzed to shed light on this. Fifth, there is need for automated real time monitoring of outbreaks for swift action and control. Notwithstanding, this antibiogram data has shown high susceptibility profiles for pathogens of public health significance.

This study is part of a collaborative network between the MRCG, and UCLA set out to improve microbiological laboratory capacity, antimicrobial surveillance, stewardship, and IPC. Diagnostic microbiological capabilities with standardized methodologies and expertise for detection and management of AMR are often lacking in many LMICs [11,37] including The Gambia. In addition, prescription guidelines are lacking in the majority of health facilities including main referral tertiary hospitals with up to half of all prescribed antimicrobials reported as inappropriate [38,39]. The global action plan on antimicrobial resistance highlights inappropriate prescribing as one of the main drivers of AMR [3]. This is further amplified by poor regulation and limited treatment options in LMICs, thus, the implementation of this antibiogram is paramount in understanding resistance dynamics to aid rational local prescribing. The antibiogram is particularly useful when the infecting organism is known prior to susceptibility patterns. This is crucial for antimicrobial prescription in our setting with only one quality assured routine diagnostic microbiology Lab. This antibiogram has resulted in the improvement of our local prescription. Being the first antibiogram analyzed according to the CLSI M39-A4 recommended guideline, a joint multistakeholder committee to improve the microbiological capacity of our diagnostic lab for antimicrobial surveillance and stewardship was formed.

## 4. Materials and Methods

### 4.1. Study Design and Setting

This study was cross-sectional descriptive analysis of the cumulative antibiogram over a period of three years (January 2016 to December 2018), using data from the Clinical Microbiology Laboratory at the Clinical Services Department (CSD) of the MRCG at LSHTM in The Gambia. A multidisciplinary team of microbiologists, physicians, epidemiologists, and IPC specialists across the collaborating institutions met over several months to review available data.

The Gambia is the smallest mainland African country in West Africa with a population of 2.1 million with high malnutrition and decreasing malaria incidence [22,40,41].

The CSD has a 42-bed capacity ward and an Outpatient Department seeing approximately 50,000 adult and pediatric patients annually. The CSD provides primary and secondary-level care to sick individuals from the surrounding population with complicated cases referred to the main tertiary government hospitals. No surgical departments, obstetrics, or Intensive Care Units are present with limited neonatal admissions. It is the only facility in the Gambia with consistent routine microbiological testing capabilities for patients with suspected invasive infections.

### 4.2. Microbiological Procedures

Samples were processed in the diagnostic microbiology laboratory, which is both Good Clinical Laboratory Practice (GCLP; 2010) certified and ISO15189 (2015) accredited. Standard microbiological processing of samples and identification procedures were regularly performed according to standard microbiological protocols. Blood culture samples were routinely collected in BD Bactec aerobic and anaerobic adult and pediatric bottles respectively. Positive bottles were cultured on blood, chocolate and MacConkey agar plates and drops put on microscopic slides for Gram staining. Growth on plates were further characterized and identified using biochemical reagents. Common normal skin flora isolates (coagulase-negative staphylococci, *Bacillus* species, *Micrococcus* spp., diphtheroids, *Propionibacterium* spp. and *Bacillus* spp. other than *B. anthracis*) from specimens of sterile origin such as blood, CSF and invasive aspirates were considered clinically non-significant and excluded from reporting unless indicated by requesting physician. First pathogens isolated for same patient was considered in the analysis. The Duke criteria for diagnosing infective endocarditis were used to determine pathogenicity of the viridans group *Streptococci* isolates. Blood and cerebrospinal fluid (CSF) samples are routinely collected for bacterial culture from patients presenting with suspected sepsis and meningitis, respectively. Microbiological data is considered community acquired as samples are collected upon presentation and long stays are rare. Data is reported and stored in a local electronic medical record system (EMRS). Selective reporting of antimicrobials is yet to be introduced and specimens were anonymized for confidentiality prior to analysis.

Urine samples were considered clinically significant when growth > 10^5^ of a single organism. Urines were primarily cultured on cysteine lactose electrolyte deficient (CLED) and incubated overnight at 37 °C, followed by appropriate pathogen identification if growth was >10^5^ CFUs. Other samples were cultured on appropriate agar plates such as blood, chocolate, MacConkey, sabouraud dextrose and Thayer-martin agar plates as indicated. The lab implements to appropriate diagnostics stewardship and body sites (wound, throat) with potential mixed microflora are only tested for antimicrobial susceptibility with the presence of a predominant organism. Isolates were identified using appropriate biochemical tests; *Enterobacterales* were identified using BioMerieux API20E, other non-enteric Gram-negative were identified using API20NE. *Staphylococcus aureus* (*S. aureus*) were identified using Staphaurex Plus and mannitol salt agar testing. *Streptococcus pneumoniae* (*S. pneumoniae*) was confirmed using optochin disc, with confirmatory testing using bile solubility. *Haemophilus influenzae* (*H. influenzae*) was identified using X and V factors and serological testing. In addition, beta-hemolytic group *Streptococci* were identified using Streptex/Wellcogen serological test. Antimicrobial susceptibility patterns were determined by Kirby-Bauer disk diffusion on Mueller-Hinton agar and zone sizes interpreted according to the relevant Clinical Laboratory Standard Institute (CLSI) guidelines on antimicrobial agents [42]. Appropriate American Type Culture Collection (ATCC) controls *E. coli* 25922, *P. aeruginosa* 29835, *S. aureus* 25923 and *S. pneumoniae* 49616 were consistently used as quality control organisms for the antibiotic susceptibility testing and reagent performance verification.

### 4.3. Analysis

Data from positive cultures for the analysis of the antibiogram were extracted from the Electronic Medical Record System (EMRS), uploaded into WHONET, cross-checked for unusual susceptibility patterns, verified as per CLSI M39 [18] and cleaned. All pathogens isolated during the period considered as individual infection episodes were included. Isolates from successive cultures from different body sites were reviewed and the first isolate included as per the CLSI guideline. Data was stratified by specimen type for blood and Urine and analyzed. Heat maps were generated to visually highlight resistance patterns and bar charts to show differences in organism frequencies among samples. Tables were generated to show cumulative antibiograms for use and implement antimicrobial prescription guideline [18].

### 4.4. Ethical Review and Approval

The study is part of an ongoing microbiological improvement project to implement local prescribing guideline and antimicrobial stewardship at the MRCG at LSHTM.

## 5. Conclusions

In conclusion, this study highlights the feasibility of implementing AMR containment strategies to improve prescribing guideline in a LMIC and has provided evidence that multistakeholder collaborative effort could be harnessed to improve antimicrobial stewardships across borders. The key involvement of the microbiology with insight into the clinical relevance of the results generated has added value into this interpretation. Working with colleagues from other parts of the globe has enabled knowledge sharing and learning from experiences of other well-established institutions actively working on AMR containment strategies. For example, other preventive strategies for reducing infection such as monitoring the hand hygiene practices of hospital staff have also been implemented. It is worth noting that the patient impact of these interventions must be evaluated in future studies. The implementation of the cumulative antibiogram using a standardized methodology for the first time is a great opportunity to further disseminate the knowledge and skills across the entire country. The information provided in this antibiogram profile has been incorporated into our clinical prescribing guideline with evidence on the appropriateness of use of WHO Access antimicrobials in our facility. However, evidence for an improvement in combination therapy is warranted locally. This has important implications for antimicrobial stewardship policy and thus, in line with global efforts to curb antimicrobial resistance. We have also demonstrated that engagement and multistakeholder collaboration can be harnessed to solve global challenges such as AMR. We have also shown how microbiology lab data can be harnessed to in AMR surveillance and stewardship in a LMIC.

## Figures and Tables

**Figure 1 antibiotics-12-00790-f001:**
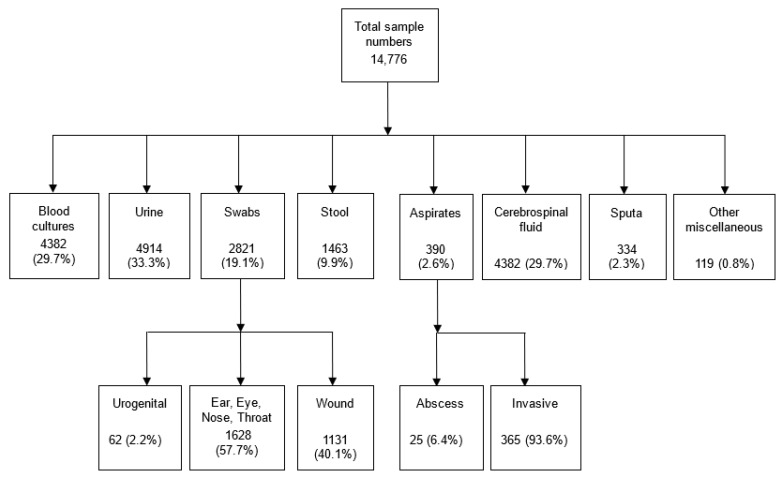
Flow chart showing distribution of the different samples.

**Figure 2 antibiotics-12-00790-f002:**
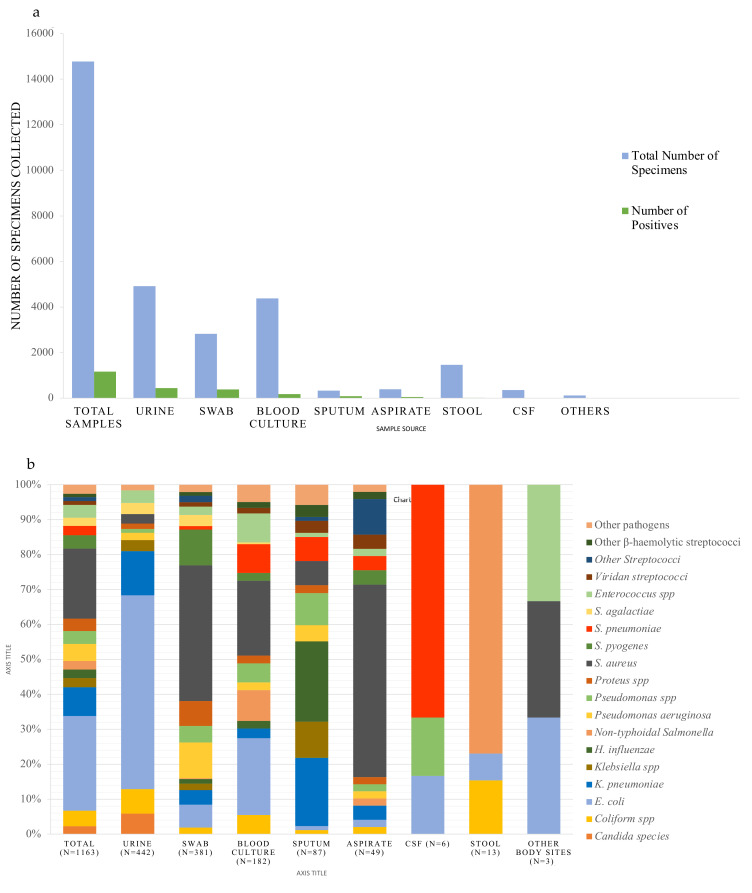
(**a**): Graph showing total number of samples received against clinically significant pathogens recovered. (**b**): Bar chart showing proportion of pathogenic organisms by sample type.

**Figure 3 antibiotics-12-00790-f003:**
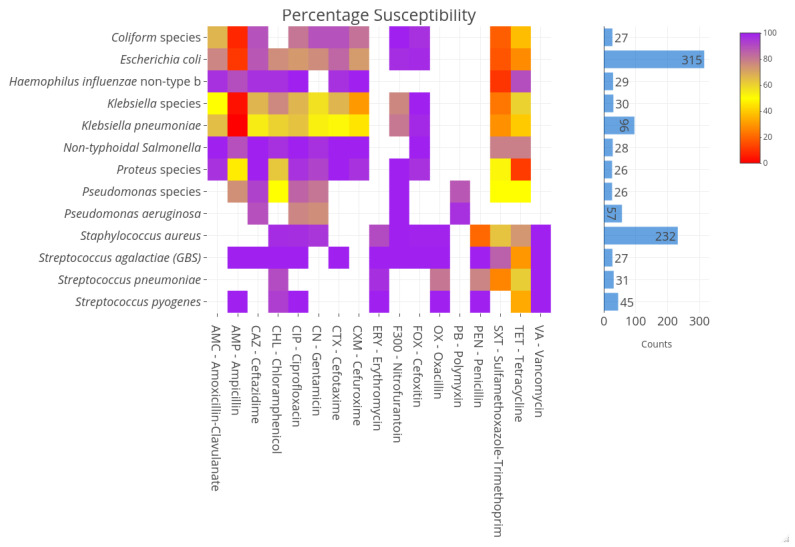
Heatmap showing the intensity of resistance patterns for the different pathogens. Purple depicting susceptibility and increasing resistance shown from yellow to red.

**Table 1 antibiotics-12-00790-t001:** Susceptibility profile for the common blood culture pathogens.

Organism	Number	% Susceptibility															
Gram-negatives		PEN	AMP	SXT	CN	CHL	TET	CIP	CXM	ERY	PB	OX	FOX	CTX	CAZ	VA	AMC
*E. coli*	40	Na	13	15	83	85	38	85	38	Na	Na	Na	97	90	88	Na	80
NTS	16 *	Na	100	88	94	100	88	100	100	Na	Na	Na	100	100	100	Na	100
*Pseudomonas* species	10 *	Na	Na	22	90	22	63	89	50	Na	Na	Na	NA	Na	100	Na	Na
**Gram-positives**																	
*S. aureus*	39	13	Na	69	97	97	74	100	Na	87	Na	100	100	Na	Na	100	Na
*S. pneumoniae*	15 *	80	Na	7	Na	93	67	Na	Na	93	Na	80	Na	Na	Na	100	Na
*Enterococcus* species	15 *	92	92	Na	Na	83	58	83	Na	58	Na	Na	Na	Na	Na	100	Na

* Less than the recommended number of 30 isolates. Key: PEN—penicillin, AMP—ampicillin, SXT—sulfamethoxazole-trimethoprim, CN—gentamicin, CHL—chloramphenicol, TET—tetracycline, CIP—ciprofloxacin, CXM—cefuroxime, ERY—erythromycin, PB—polymyxin B, OX, FOX—cefoxitin, CTX—cefotaxime, CAZ—ceftazidime, VA—vancomycin, AMC—amoxicillin-clavulanate, Na—Not applicable.

**Table 2 antibiotics-12-00790-t002:** Susceptibility profile for the urine pathogens.

Organism	Number	% Susceptibility																
Gram-negatives		PEN	AMP	OB	OX	SXT	CN	TET	F300	CIP	ERY	CXM	CAZ	CTX	FOX	AMC	PB	VA
*E coli*	245	Na	11	Na	Na	16	72	26	97	69	Na	78	88	84	98	77	Na	Na
*K. pneumoniae*	56	Na	0	Na	Na	16	46	33	80	59	Na	29	41	40	96	54	Na	Na
Coliform species	31	Na	7	Na	Na	23	87	38	100	83	Na	74	80	87	87	61	Na	Na
*Klebsiella* species	14 *	Na	0	Na	Na	7	36	57	79	57	Na	36	50	50	100	29	Na	Na
**Gram-positives**																		
*Candida species*	26 *	Na	Na	Na	Na	Na	Na	Na	Na	Na	Na	Na	Na	Na	Na	Na	Na	Na
*Enterococcus* species	16 *	94	88	Na	Na	Na	Na	13	94	75	81	Na	Na	Na	Na	Na	Na	94
*S. agalactiae*	14 *	100	100	Na	Na	86	Na	7	100	100	100	Na	Na	Na	Na	Na	Na	100
*S. aureus*	12 *	58	Na	100	100	58	100	92	100	100	100	NA	Na	Na	100	Na	Na	100

* Less than the recommended number of 30 isolates.

## Data Availability

Not applicable.

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
