# Peer review of "Using an Antibiogram Profile to Improve Infection Control and Rational Antimicrobial Therapy in an Urban Hospital in The Gambia, Strategies and Lessons for Low- and Middle-Income Countries"

_antibiotics, 2023, doi:10.3390/antibiotics12040790_

Round 1
Reviewer 1 Report
Dear all,
The manuscript entitled “Implementing an antibiogram profile to aid rational antimicrobial therapy and improving infection prevention in an urban hospital in The Gambia; strategies and lessons for low-income and middle-income countries.” shows the pathogen diversity and antibiotic susceptibility of bacteria that caused infections in an urban hospital in Gambia over the course of three years. Even though generally well-written, the impact of this antibiogram implementation is very superficially discussed: the authors claim that these results could be used to inform therapeutic guidelines but, in my opinion, the practical implications of this study are not thoroughly explored. This, together with important formatting issues, drives me to recommend acceptance after major revisions are conducted.
General comments:
-
References need to be appropriately and coherently formatted.
-
The title should not contain semi-colons or full stops.
Detailed comments:
-
Line 30: A full stop is missing after the word “disease”.
-
Line 32: The word “and” should be placed before the last enumerated antibiotic or mixture of antibiotics.
-
Lines 42-43: The sentence “with sub-Saharan Africa (sSA) disproportionately affected” should be rephrased to “particularly in sub-Saharan Africa (sSA) populations that are disproportionately affected by this phenomenon compared to other regions of the globe”, or an equivalent sentence. Otherwise, it does not make sense.
-
Figure 1: Please remove the title and add abbreviations for CSF and EENT. Also, there are spaces missing between the number of samples and the percentage, e.g. 4382(29.7%) should be formatted to 4382 (29.7%).
-
The criteria for clinical significance are not presented except for urine samples. Please introduce this information in the manuscript.
-
Pathogen names should be written in italic throughout the manuscript.
-
Figure 2: Please remove the heading and keep only A and B for Graph identification. Also, the axes of Figure 2b are not appropriately labeled.
-
Tables 1 and 2: Please provide abbreviature meanings for all antibiotics, as well as for “Na”.
-
Table 1 should be placed after the first citation (section 2.2).
-
Figure 3: Pathogen names should be written in italic. Antibiotic abbreviations should be added to the Figure legend. Also, please delete the icons on the top right corner of the figure, and visually add labels for the purple-red gradient.
-
Line 225: “The emerging penicillin resistance warrant…” Warrant should be replaced by warrants.
-
Line 253: “In additions” should be replaced with “In addition”.
-
Line 257: “...resource limited…” should instead read “...resource-limited…”.
-
Lines 246-248: I don’t see how “This study highlights the feasibility of implementing AMR containment strategies to improve prescribing guideline in a LMIC and has provided evidence that multistake-holder collaborative effort could improve antimicrobial stewardships across borders” by simply providing data on pathogen profiles and antimicrobial susceptibilities found in locally-collected samples. Please broaden your discussion to clarify this point.
-
I feel that, overall, the implications of these results are very superficially discussed. Sentences such as “In the advent of AMR, data on local context could be used to inform therapeutic guidelines and improve prescribing and impact on stewardship.” do not clarify exactly how the results presented in this manuscript will “aid rational antimicrobial therapy” and improvement of “infection prevention” that is announced in the title. One thing is sorting out the antibiogram, another is using it to remodel antimicrobial therapy prescription policies. Please take this into account when reframing your discussion.
Kind regards.
Author Response
All comments have been addressed in the file attached.

Reviewer 2 Report
Exploring the bacterial resistance pattern in local health facilities is a cornerstone in constructing a national robust antibiogram, the authors did well to tackle this point, however, they need more effort to present their findings to be of real value as guidance for empiric use of antimicrobials in their health facility.
Title: The article title is wordy, I suggest summarizing it. Ex: “Using an antibiogram profile to improve infection control and rational antimicrobial therapy in an urban hospital in The Gambia; strategies and lessons for low- and middle-income countries.”
Results
1) The results section comes after the methodology section.
2) Line 84-87: out of 14,776 different specimens, only 1163 were considered clinically significant cultures. The author may need to explain how they excluded the remaining specimens, such as whether they had no bacterial growth, were duplicate samples with the same organisms and susceptibility patterns, represented contamination or colonization, or had fewer than 30 positive microbiological cultures.
3) I suggest reformatting the flowchart in figure (1) based on the inclusion and exclusion criteria of microbiological specimens.
4) Line 90-134: the vast majority of the specimens were excluded from the antibiogram construction, please refer to comment no.2.
5) This section should include information on the following demographics, the patient's age, gender, and nursing unit (such as a surgical ward, medical unit, or critical care unit).
6) Interpretable antibiograms should be presented as separate tables for:
ü Gram-positive, Gram-positive, anaerobes.
ü Hospital-acquired Vs. community-acquired infections.
ü Source of the sample, and ward location (Surgical, medical, critical care)
ü Emerging Resistance Trends
8) Recommendations regarding the empiric use of antimicrobials in various situations will be added value to the antibiogram, I advise the authors to include tailored recommendations/interventions to use the provided information as guidance for empiric antimicrobials prescribing to mitigate unnecessary use and emergence of resistance.
Methods
9) The method section fails to recognize colonization or infection, the authors need to identify how they deal with this aspect when including the cultures for descriptive analysis.
10) Identifying hospital- and community-acquired infections must be included in a cumulative robust antibiogram; it is suggested that this information be included in the manuscript.
11) The manuscript should provide a brief about the phenotypic/genotypic classification of the pathogens, MIC values, and breakpoints for antimicrobial therapies.
12) The CLSI-2018-M100-S28 guidelines explicitly clarify the inclusion and exclusion of microbiological cultures in the analysis; this should be explained in the method section.
General comments
1) This manuscript lacks a discussion section, which I believe is essential for interpreting the results, comparing them to other studies, and making specific recommendations.
2) The conclusion is brief and offers no recommendations based on the findings.
3) English language and grammar review are required.
4) To give readers a clear message, the graphic presentation and tables need to be restructured.
5) The work needs rearrangement of article sectors and rephrasing.
Author Response
All comments have been addressed.

Round 2
Reviewer 1 Report
Dear all,
Please note that the axes of Figure 2b have not been corrected. Otherwise, I see that the manuscript was substantially improved, and thus I am supportive of acceptance.
Kind regards
Reviewer 2 Report
None of the cycle 1 comments were properly addressed.
